# CRITIQUE ABILITY OF LARGE LANGUAGE MODELS

## ABSTRACT

Critical thinking is essential for rational decision-making and problem-solving. This skill hinges on the ability to provide precise and reasoned critiques and is a hallmark of human intelligence. In the era of large language models (LLMs), this study explores the ability of LLMs to deliver accurate critiques across various tasks. We are interested in this topic as a capable critic model could not only serve as a reliable evaluator, but also as a source of supervised signals for model tuning. Particularly, if a model can self-critique, it has the potential for autonomous self-improvement. To examine this, we introduce a unified evaluation framework for assessing the critique abilities of LLMs. We develop a benchmark called CRITICBENCH, which comprises 3K high-quality natural language queries and corresponding model responses; and annotate the correctness of these responses. The benchmark cover tasks such as math problem-solving, code completion, and question answering. We evaluate multiple LLMs on the collected dataset and our analysis reveals several noteworthy insights: (1) Critique is generally challenging for most LLMs, and this capability often emerges only when models are sufficiently large. (2) In particular, self-critique is especially difficult. Even top-performing LLMs struggle to achieve satisfactory performance. (3) Models tend to have lower critique accuracy on problems where they are most uncertain. To this end, we introduce a simple yet effective baseline named *self-check*, which leverages self-critique to improve task performance for various models. We hope this study serves as an initial exploration into understanding the critique abilities of LLMs, and aims to inform future research, including the development of more proficient critic models and the application of critiques across diverse tasks.

## 1 INTRODUCTION

> *"Self-criticism is an art not many are qualified to practice." — Joyce Carol Oates*

Large language models (LLMs) have demonstrated impressive capacities in a wide range of tasks (Google et al., 2023; OpenAI, 2023). Consequently, the evaluation of LLMs has shifted focus from basic sentence coherence to more advanced capabilities, *e.g.*, knowledge acquisition and logical reasoning (Hendrycks et al., 2021; BIG-Bench authors, 2023). One capability that is overlooked in current evaluation frameworks is the ability of *critical thinking*, which is an important hallmark of human intelligence that requires logic, reasoning, and knowledge. This ability ensures that LLMs can provide precise and reasoned critiques towards model responses. A model with robust critique ability can identify potential misinformation, errors or context misalignment in model outputs, thereby showing their specific shortcomings that can serve as a feedback for improvement. While recent studies have used LLMs for various forms of critique across diverse applications (Madaan et al., 2023; Saunders et al., 2022; Shinn et al., 2023), they primarily focus on advancing the state of the art for specific tasks instead of providing a comprehensive assessment of critique ability.

To address this gap, we propose a standardized benchmark **CRITICBENCH** to assess the critique abilities of LLMs in diverse tasks. We define a model's critique ability as "*the capacity to identify flaws in model responses to queries*". Figure 1 provides an example of a flaw in the response to a query, and how it is identified by a critique. The benchmark consists of query-response-judgment triplets. During evaluation, we always prompt a model to perform a chain-of-thought analysis to identify flaws and explain the reason; and then provide a final judgment on the response's correctness. Comparing this judgment to ground-truth labels allows us to *explicitly* evaluate a model's critique accuracy and *implicitly* assess its analytical process toward an accurate judgment.

> **Query:** Heather's razors come 4 to a pack and cost $4.00 a pack. They are currently on sale for buy one get one free. She also has a $2.00 coupon. How much will each individual razor cost, in cents, after the discount if she buys 2 packs of razors?
> **Response:** She bought 2 packs. Each pack has 4 razors, so in total she has 2 * 4 = 8 razors. Each pack costs $4. She has a coupon of $2. So she paid 2 * $4 - $2 = $6. Each razor costs $6/8 = $0.75. The answer is 75.
> **Certainty Score:** 0.11
> **Correctness Rate:** 0.93
> - - - - - - - - - - - - - - - - - - - - - - - - - - - - - - - - - - - - - - - - - -
> **Analysis:** The student correctly calculated the total cost of 2 packs of razors with the coupon. However, they didn't consider the fact that the second pack is free. So, Heather only paid for one pack, not two. Therefore, the cost per razor is $2 / 8 = $0.25. The answer is incorrect.
> **Judgment:** incorrect.

Figure 1: An example from CRITICBENCH is presented. The query originates from GSM8K (Cobbe et al., 2021), and the response is generated by PaLM-2-L (Google et al., 2023). A flaw in the response is highlighted in red. The model shows low confidence in this query, as evidenced by a certainty score of only 0.11. Below the dashed line, a critique is generated by few-shot prompting PaLM-2-L. It successfully identifies the flaw in the response and makes an accurate judgment. As the policy model and critic model are the same, this example also serves as an instance of *self-critique*.

To construct CRITICBENCH (Section 3), we gather natural language queries from multiple scientific benchmarks, covering tasks like math problem-solving (Cobbe et al., 2021), code completion (Chen et al., 2021), and question answering (Lin et al., 2021). We employ PaLM-2 models (Google et al., 2023) of various sizes to generate responses, which are then annotated for correctness. To ensure data quality, a complexity-based selection strategy (Fu et al., 2023b) is used to identify high-quality responses among the candidates. Furthermore, to select queries of suitable difficulty, we introduce an auxiliary metric that quantifies a model's certainty regarding a query. Such a metric can help select queries that poses a moderate level of challenge to models. As a result, we collect 3K high-quality examples from an initial pool of 780K candidates to form the benchmark mixture. This data collection method is both scalable and generalizable, requiring no extra human intervention and suitable for a variety of tasks.

Given CRITICBENCH, we can now analyze the critique abilities of LLMs (Section 4). There are specific aspects that particularly interest us. First, critique inherently involves logic, reasoning, and knowledge, making it a complex process even for humans. Therefore, it is not clear how well LLMs can emulate this capability. It is possible that critique ability is yet another emergent ability, *i.e.*, ability not present in smaller-scale models that are present in larger-scale models (Jang, 2023). Investigating how critique ability scales with model size could offer insights into model size selection and whether fine-tuning is needed for smaller models (Section 4.1). Additionally, *self-critique*, *i.e.*, when a model critiques its own outputs, is a format of critique of particular interest to us, as it is relevant to a model's potential for self-improvement (Section 4.2). Finally, we are also interested in what types of queries pose more challenges for LLMs to critique (Section 4.3).

To investigate these aspects, we evaluate various widely-used LLMs on CRITICBENCH and reveal several intriguing findings: (1) Critique tasks pose a considerable challenge for LLMs. Only large-scale models exhibit performance with a notable difference from a random guess baseline, indicating that the capacity for critique serves as an emergent indicator of a capable LLM. (2) Self-critique, *i.e.*, a model critiquing its own output, is particularly difficult. Even the strongest LLMs struggle to achieve satisfactory performance. (3) A challenging query is not only difficult for LLMs to directly answer correctly, but also poses a challenge in assessing an answer's correctness to that query.

To this end, we also propose a simple yet effective baseline called *self-check* (Section 5). The basic idea is to prompt the model to confirm the accuracy of their generated answers by self-critique before presenting them. The method consistently enhances the baseline performance (Wang et al., 2023) on math word problems across multiple models, achieving an average of 9.55% error reduction rate, which demonstrates the potential utility of critiques from LLMs.

Our contributions are three-fold:

- **New Benchmark** CRITICBENCH is the first benchmark that comprehensively assesses the critique abilities of LLMs across diverse tasks and scenarios, which fills a gap in the current LLM evaluation framework by introducing this important ability.

- **New Findings** Our findings on CRITICBENCH underscore the nuances and depth of LLM's critique abilities (Section 4). These revelations enhance our understanding of the inherent complexities in LLMs and emphasize the need for advanced training and evaluation techniques.
- **New Capacity** The proposed *self-check* method (Section 5) not only advances the performance on math word problems over the baseline, but also indicates the new capacity of critique ability with LLMs, which is a fruitful avenue for LLM's self-improvement strategies.

## 2 DEFINITION OF CRITIQUE ABILITY

The concept of *critique* has diverse interpretations and is often applied informally in everyday contexts. Recent research employs large language models to offer critiques across multiple applications (Madaan et al., 2023; Paul et al., 2023; Saunders et al., 2022; Shinn et al., 2023), resulting in varying formats and requirements for their "critiques". These studies primarily aim to enhance performance in specific tasks, neglecting to clarify the meaning of the term critique. In this paper, we consider the definition of a language model's critique ability as

*the capacity to identify flaws in model responses to queries.*

These flaws can differ depending on the task, ranging from incorrect reasoning or calculation in mathematical problems to syntax errors in code completion.

When a model self-assesses its own outputs, we term this as *self-critique*, a notion that particularly intrigues us. If models can engage in self-critique and reflection, they can potentially do self-improvement, requiring minimal human intervention. On the risky side, this autonomy also raises concerns about reduced human oversight (Bowman et al., 2022). Yet we posit that self-critique may still remain a challenging capability for large language models, as a flaw-aware model would logically not produce faulty output in the first place (Saunders et al., 2022).

## 3 CONSTRUCTION OF CRITICBENCH

As discussed in Section 2, prior research employs large language models to offer critiques, yet requires particular process and formats to meet their task-specific objectives. Currently, there is no standard or generalizable way to assess the critique abilities of language models across diverse tasks. This section proposes CRITICBENCH, a unified, standardized evaluation framework to tackle this issue. The framework aims to fulfill three criteria:

- **Scalability** Given the broad range of tasks already established within the community, and the anticipation of more to emerge, a scalable data collection method is essential. The method should minimize human annotation efforts and ideally be fully autonomous.
- **Generalizability** The framework should be task-agnostic, capable of generalizing across various tasks and domains.
- **Quality** We believe quality matters more than quantity. When volume of data is substantial, we prioritize selecting those that most effectively differentiate between stronger and weaker models.

The following subsections illustrate the detailed construction process. Specifically, Section 3.1 presents the initial data generation on three different tasks, where we get the collection of query-response-judgment triplets as shown in Figure 1. Section 3.2 then shows how to select data based on the initial collection to guarantee the quality of responses and queries.

### 3.1 DATA GENERATION

For the tasks of interest, we begin by employing existing scientific datasets from relevant domains. These datasets are expected to include queries that large language models, which here we refer to as *generators*, aim to respond.

To ensure scalability, it is essential to have an automated approach for assessing the correctness of a model's responses. Classification tasks naturally meet this criterion, as model outputs can be automatically compared to ground-truth labels. Similarly, tasks that involve auto-verifiable answers also comply; for instance, in code completion tasks with unit tests available, the validity of the generated

code can be confirmed by passing all tests. For free-form generation tasks such as summarization and translation, assessing the quality of a response remains non-trivial. However, recent advances in LLM-based automated evaluation for generation tasks mitigate this issue to some extent, enabling the assessment without human intervention (Liu et al., 2023).

While not exhaustive, these already cover a significant range of tasks and domains. We acknowledge the limitations in some auto-assessment approaches, especially for generation tasks. Improving the reliability of these automated evaluation methods, however, is beyond the scope of this paper.

We employ five different sizes of PaLM-2 models (Google et al., 2023) as our generators. These models are pretrained solely for next-token prediction and do not undergo supervised fine-tuning or reinforcement learning from human feedback. For coding-related tasks, apart from the standard PaLM-2 models, we also employ the specialized PaLM-2-S* variant. The latter is obtained through continual training of PaLM-2-S on a data mixture enriched with code-heavy corpus.

**Query Collection**   We extract queries from three datasets: GSM8K (Cobbe et al., 2021), HumanEval (Chen et al., 2021), and TruthfulQA (Lin et al., 2021), covering the tasks of math-problem solving, code completion and question answering. For datasets with distinct training and test splits, we use the test data; for datasets intended only for evaluation, all examples are used. Detailed considerations and rationale behind the selection of these datasets are provided in Appendix B.

**Response Generation**   We sample $k$ responses for each query, with $k = 64$ for GSM8K and TruthfulQA, and $k = 100$ for HumanEval. In the case of TruthfulQA, we employ its multiple-choice variation to facilitate autonomous answer annotation. After filtering out invalid outputs such as empty ones, we collect a total of 780K responses as an initial pool of candidates.

**Annotation for Correctness**   For GSM8K, we assess answer correctness by comparing its numeric equality to the ground truth, as described by Lewkowycz et al. (2022). For HumanEval, correctness is determined by the passage of provided unit tests. For TruthfulQA, we utilize its classification format, judging correctness based on a match with the ground-truth label.

More details on hyper-parameter settings and prompt templates are available in Appendix C.

## 3.2   DATA SELECTION

Many existing evaluation benchmarks for large language models suffer from insufficient differentiability, *i.e.*, both stronger and weaker models yield similar performance (Fu, 2023). This issue likely arises from the presence of either overly simple or exceedingly difficult examples in the benchmarks. Such examples are less valuable for evaluation and can undermine the utility of the benchmarks when average scores are calculated, leading to indistinguishable outcomes. To address the issue, we introduce various filtering strategies aimed at selecting high-quality and differentiable examples.

### 3.2.1   HIGH-QUALITY RESPONSE SELECTION

Initially, we can narrow the example set from 780K to 15K by sampling one correct and one incorrect response for each query and generator. While random uniform sampling is the the most straightforward strategy, it risks including examples with obvious errors, which offer little evaluative value. To mitigate this, for the incorrect responses we focus on sampling *convincing wrong-answers* (Lightman et al., 2023) that are more likely to fool the models. In cases suitable for majority voting, we identify the most frequent incorrect answer for each query, and then sample from responses containing this answer. For coding tasks where majority voting is not applicable, we sample from responses that pass the most unit tests, indicating that it is mostly correct but fails in certain corner cases.

To further enhance data quality, we employ the *complexity-based* sample selection strategy (Fu et al., 2023b) for tasks that require chain-of-thought reasoning. Specifically, we opt for responses that involve more reasoning steps, as this is positively correlated with higher accuracy (Fu et al., 2023b). This approach is beneficial for sampling both types of responses. For correct ones, it minimizes the likelihood of false positives; for incorrect ones, it yields more convincing responses that pose greater challenges in flaw detection for weaker models.

Lastly, as many tasks are challenging and may require emergent abilities (Wei et al., 2022a) to perform well, smaller models generally underperform and produce lower-quality responses compared

to larger ones. We include data from these smaller models only for analyzing self-critique abilities; they are excluded from the final evaluation benchmarks.

### 3.2.2 CERTAINTY-BASED QUERY SELECTION

Thus far, our focus has been on choosing responses with higher quality and likelihood of accuracy. We now shift our focus to the quality of queries. Not all queries are equally valuable: trivial queries that models easily answer correctly are generally less valuable, whereas queries consistently answered incorrectly may either be too complex for LLMs or suffer from wrong "golden" labels.

To minimize the presence of such queries in our benchmark, we introduce two metrics to quantify the levels of certainty when models answer a query: the **certainty score** and **correctness rate**. We will use these metrics to help us select queries of moderate difficulty.

The metrics draw inspiration from the majority voting mechanism in the *self-consistency* approach (Wang et al., 2023), which functions by generating multiple candidate outputs for a query, and then aggregating them using a majority voting procedure to select the most commonly occurring answer. Observing that different majority votes, even those resulting in the same outcome, can indicate vastly different levels of certainty. To illustrate, consider a voting situation with $100$ candidates where: (i) all candidates are $x$; and (ii) $51$ candidates are $x$ and $49$ are $y$. Although both situations result in a majority vote for $x$, the level of certainty varies significantly: the former situation denotes a high degree of confidence, whereas the latter reflects a considerable level of uncertainty.

Motivated by the observations above, we propose the following method to measure levels of uncertainty in language model responses. Suppose we prompt a language model $\text{LM} : P(a|q)$ with a query $q$ and sample a bag of $k$ answers $\mathbb{A} = \{a_i\}_{i=1}^{k}$, where $a_i \sim P(a|q)$. We denote the most and the second most frequent answers among these $k$ responses as $a^{(1)}$ and $a^{(2)}$, respectively. Uncertainty is measured by the frequency ratio of $a^{(2)}$ to $a^{(1)}$, where a larger ratio indicates a higher level of uncertainty. We term this ratio as *uncertainty rate*. An uncertainty rate of $1$ — where the two most frequent answers appear with equal frequency — indicates extremely high model uncertainty. Conversely, an uncertainty rate of $0$, implying that $a^{(2)} = 0$, suggests that all responses are consistent, indicating the model's strong confidence in its answer.

Formally, we use $f_{\mathbb{A}}(a) = \sum_{a_i \in \mathbb{A}} \mathbf{1}_{\text{condition}}(a_i = a)$ to denote the frequency of an answer $a$ among a bag of responses $\mathbb{A}$ and $\text{mode}(\mathbb{A}) = \arg\max_a f_{\mathbb{A}}(a)$ to denote the *mode*, *i.e.*, the most frequently occurring item , of $\mathbb{A}$. The uncertainty rate over model responses $\mathbb{A}$ to the query $q$ is then defined as $\text{UR}_{\text{LM}}(q; k) = \frac{f_{\mathbb{A}}(\text{mode}(\mathbb{A} \backslash \mathbb{A}^{(1)}))}{f_{\mathbb{A}}(\text{mode}(\mathbb{A}))}$, where $\mathbb{A}^{(1)} = \{a \mid a = \text{mode}(\mathbb{A}), a \in \mathbb{A}\}$ represents the most frequent responses in $\mathbb{A}$. For the sake of conciseness and readability in our subsequent discussion, we also define a metric by the negative logarithm of the uncertainty rate to measure model certainty, represented as $\text{Certainty}_{\text{LM}}(q; k) = -\log(\text{UR}_{\text{LM}}(q; k))$, where a larger value indicates a higher level of certainty. We term it as the **certainty score**.

In cases where the expected correct answer to a query is available, such as during model evaluation on a test dataset, the definitions above can be slightly adapted to introduce a new metric called **correctness rate**. This metric is defined as the frequency ratio of the correct answer to the most common wrong answer: $\text{CR}_{\text{LM}}(q; k) = \frac{f_{\mathbb{A}}(a^{(\text{e})})}{f_{\mathbb{A}}(\text{mode}(\mathbb{A}_{\text{wrong}}))}$, where $a^{(\text{e})}$ denotes the expected answer and $\mathbb{A}_{\text{wrong}} = \{a \mid a \neq a^{(\text{e})}, a \in \mathbb{A}\}$ denotes the incorrect responses. Using self-consistency, the model votes a correct answer when the correctness rate exceeds $1$, and conversely, it produces an incorrect answer when the rate falls below $1$. In addition, as the rate approaches $1$, the model exhibits increasing levels of uncertainty regarding the answer, no matter if it is correct or not. This metric naturally reflects the difficulty of a query to the model.

We present a simple case study to intuitively demonstrate the properties of our proposed metrics. We evaluate PaLM-2-S (Google et al., 2023) on GSM8K (Cobbe et al., 2021) using a 64-path self-consistency. The relationship between model certainty, correctness rate (CR), and model accuracy is depicted in Figure 2.

Figure 2a displays the correlation between model certainty and correctness rate (CR). Test examples with lower CR present greater challenges to models. As evidenced in the figure, lower certainty correlates with more low-CR examples, leading to more incorrect predictions. As certainty increases,

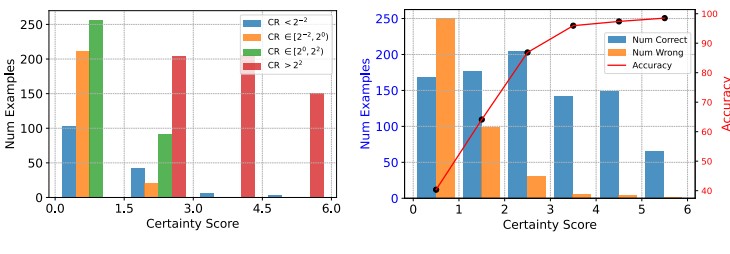

(a) Relation of certainty to CR.      (b) Relation of certainty to accuracy.

Figure 2: Certainty of PaLM-2-S on GSM8K: Relation to correctness rate (CR) and accuracy; based on the 8-shot chain-of-thought prompt from Wei et al. (2022b) and a 64-path self-consistency.

the instances of low CR diminish, resulting in higher accuracy. Figure 2b illustrates the correlation between model certainty and accuracy in a more straightforward way. As the certainty level rises, the proportion of incorrect predictions markedly decreases, signifying increased accuracy.

We now adopt a *certainty-based* sample selection strategy. We calculate the correctness rate for each query, selecting those with a CR close to 1. This suggests that models exhibit considerable hesitation and uncertainty when responding to these queries, indicating a moderate level of difficulty that is neither excessively simple (CR $\to +\infty$) nor overly challenging (CR $\to 0$). For coding tasks, where certainty metrics cannot be computed, we use the ratio of correct to incorrect answers as a surrogate for CR. Moreover, due to the limited size of HumanEval, we only exclude the simpler queries with a CR $> 1$, and retain the challenging examples. We will analyze the correlation between critique ability and model certainty for queries in Section 4.3.

Detailed implementation of each stage in data selection can be found in Appendix D.

**Final Data Formulation**    To this end, we could further narrow the benchmark dataset to 3K high-quality, differentiable examples, with 1K for each original dataset. The resulting subsets are named as Critic-GSM8K, Critic-HumanEval, and Critic-TruthfulQA, and their mixture is referred to as **CRITICBENCH**. We provide the data statistics and examples in Appendix E. As our data collection method is scalable and generalizable across tasks, we view the construction of CRITICBENCH as a continuous effort. This paper serves as an initial step, presenting three representative datasets. We hope to extend the mixture to cover more tasks and scenarios in future work.

## 4    PROPERTIES OF CRITIQUE ABILITY

In this section, we conduct our analysis of the critique ability of large language models on CRIT-ICBENCH. We focus primarily on the following three aspects: (1) how critique ability scales with model size (Section 4.1); (2) models' self-critique ability (Section 4.2); and (3) the correlation between critique ability and models' certainty in response to a query (Section 4.3).

For each query-response pair in the dataset, we employ few-shot prompting to instruct models to first conduct a chain-of-thought analysis to identify any flaws in the response and explain the reason; and subsequently issue a judgment on the response's correctness. In evaluation, we focus solely on the accuracy of this final judgment, disregarding the correctness of the intermediate analysis. As empirical evidence has shown a strong correlation between the accuracy of intermediate chain-of-thought and the final answer (Wei et al., 2022b; Lewkowycz et al., 2022; Fu et al., 2023a), we can use the final judgment accuracy as a proxy for the model's critique analysis capability. Details about the evaluation settings can be found in Appendix F.

### 4.1    SCALING LAW

Jang (2023) posits that critique ability may be an emergent ability (Wei et al., 2022a) that only emerges at certain scales of model size. We emphasize that it is better to seek an answer to this hypothesis before directing our efforts toward the applications of critiques. For a critic model to successfully improve the performance of specific tasks, it must possess at least moderate effectiveness. It is possible that the critique ability of smaller models is as futile as a random guess,

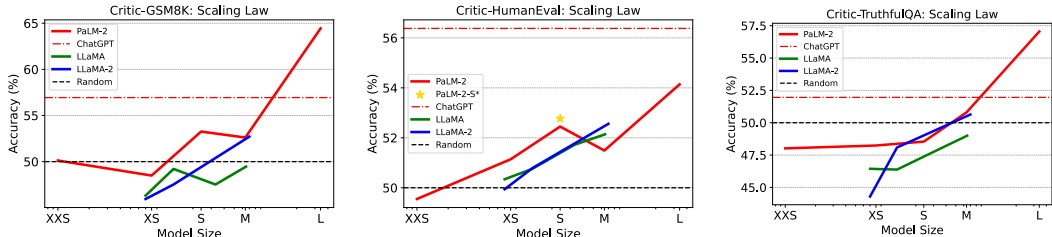

Figure 3: Scaling law of critique ability: Following Google et al. (2023), we use T-shirt size notations to denote model sizes. All medium-sized or smaller models exhibit poor performance on all tasks, akin to random guessing. Critic-HumanEval poses a great challenge for all models.

rendering them incapable for downstream applications. A study of the *scaling law* of critique ability could provide us insights into the appropriate model size selection and whether fine-tuning should be considered for smaller models.

We evaluate multiple widely-used LLM families available in various sizes on CRITICBENCH, including PaLM-2 (Google et al., 2023), LLaMA (Touvron et al., 2023a), LLaMA-2 (Touvron et al., 2023b), and ChatGPT (OpenAI, 2023). Figure 3 illustrates the scaling behavior of their critique abilities. The results for ChatGPT are not directly comparable to those of other models because its size is not disclosed and it undergoes instruction-tuning, whereas the others are all pretrained models. We include it here solely for reference purpose. On Critic-GSM8K and Critic-TruthfulQA, all models of medium size or smaller exhibit poor performance, akin to random guessing. Only PaLM-2-L demonstrates non-trivial better results. On Critic-HumanEval, all models perform poorly; even the strongest pretrained model, PaLM-2-L, only achieves an accuracy score of merely $54.14\%$, which is just marginally better than a random guess. This is somewhat anticipated, as evaluating the correctness of a code snippet without execution is often challenging even for expert software engineers. It is likely to gain a notable improvement when augmented by a code interpreter tool. Thus, the benchmark also serves as an ideal testbed to assess LLMs' tool-use capability.

The observed scaling law supports the emergent ability hypothesis by Jang (2023). It suggests that the ability of critique is yet another key indicator of a strong large language model.

## 4.2 SELF-CRITIQUE ABILITY

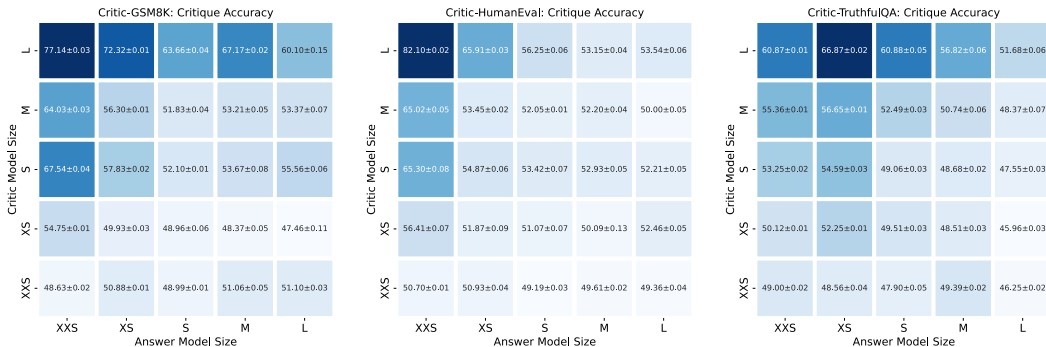

Figure 4: The accuracy of differently-sized critic models in critiquing answers produced by differently-sized policy models. For instance, the top-left cells indicate the accuracy of PaLM-2-L in critiquing answers from PaLM-2-XXS.

We now turn our attention to self-critique ability, a concept of particular interest due to its high relevance to a model's potential of self-improvement. Figure 4 demonstrates the critique performance of various sizes of critic models in evaluating answers produced by different-sized policy models. The diagonal lines spanning from the lower left to the upper right represent the models' self-critique accuracy, and correspond to the curves in Figure 5.

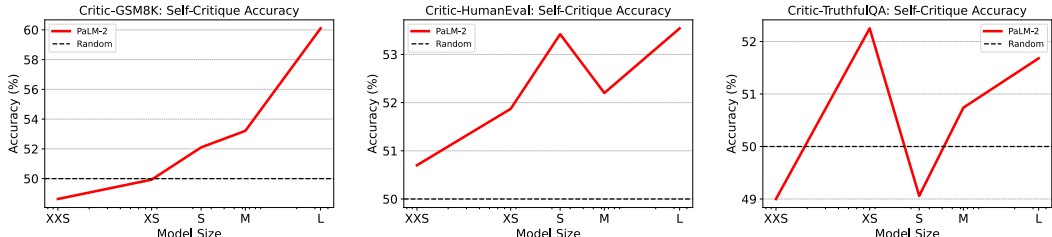

Figure 5: Self-critique accuracy of PaLM-2 models: On Critic-GSM8K, larger models demonstrate better self-critique ability. On the other two tasks, all models perform poorly.

The scaling behavior varies across different subsets. It is unsurprising that models of all sizes struggle on Critic-HumanEval due to its challenging nature. On Critic-GSM8K, larger models display better self-critique ability. On Critic-TruthfulQA, however, models perform similarly to random guessing regardless of model size. We hypothesize the disparity is due to the underlying reasons of a model answering incorrectly to queries. For TruthfulQA, the wrong answers largely stem from false beliefs or misconceptions in models, which would also lead to critique failures. In contrast, for the math queries in GSM8K, incorrect responses primarily result from reasoning or computational flaws, which are likely detectable upon a double check through self-critiquing.

Another finding is larger models are generally good at critiquing responses generated by smaller models. The outcome aligns with the expectation that smaller models are more prone to more obvious errors, which are easier caught by larger and more capable models.

## 4.3 CORRELATION TO CERTAINTY

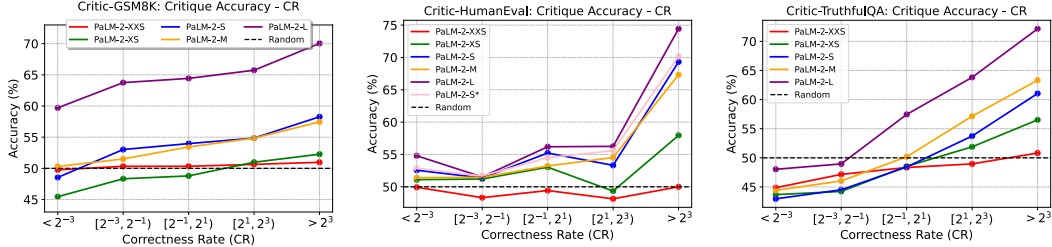

Figure 6: Relation to correctness rate (CR).

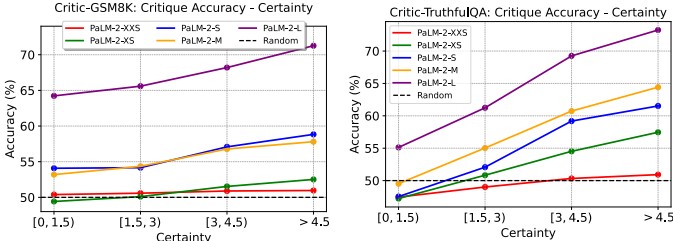

Figure 7: Relation to certainty score.

In Section 3.2.2, we introduce the use of certainty metrics to select queries of appropriate difficulty. While the metrics do reflect the challenge of answering a query, one may argue that it does not directly translate to the difficulty of critiquing an answer to that query. To address this, we examine the correlation between critique accuracy and model certainty for a query. We evaluate PaLM-2 models on the benchmarks without applying certainty-based selection. Figures 6 and 7 display the correlation between critique ability, correctness rate, and certainty score. Note that for Critic-

HumanEval, we cannot compute the certainty score because it is not applicable to majority voting for code snippets. Additionally, the correctness rate is calculated differently as detailed in Section 3.2.2.

We observe a clear positive correlation between model certainty and critique accuracy. This suggests that a challenging query is not only difficult for LLMs to directly answer correctly, but also poses a challenge in evaluating an answer's correctness to the query. Consequently, the proposed certainty metrics serve as valuable criteria for data selection.

## 5    NEW CAPACITY WITH CRITIQUE: SELF-CONSISTENCY WITH SELF-CHECK

To explore the new capacity with critique ability, we would like to introduce a straightforward yet effective baseline to demonstrate the potential of leveraging the critique ability to improve model performance. The idea is intuitive: drawing a parallel to humans participating in a contest — where they typically check their most uncertain answers before submission to identify and correct mistakes — we suggest a similar process can be emulated in language models. This can be accomplished by prompting the models to confirm the accuracy of their generated answers before presenting them.

To achieve this, we introduce a *self-check* filtering on top of the *self-consistency* method (Wang et al., 2023), abbreviated as $\mathbf{SC^2}$. Assume with appropriate prompting, the language model functions as an answer-critiquing model $V(a) \in \{0, 1\}$, which serves as a binary indicator for the correctness of an answer $a$ relative to its query $q$. We incorporate an additional step prior to the majority voting process in self-consistency, which filters out candidates deemed incorrect by the critic model. Specifically, for a set of $k$ generated candidate answers $\mathbb{A}$ to a given query, the critic model selects those identified as correct, denoted by $\mathbb{A}_{sc} = \{a \mid V(a) = 1, a \in \mathbb{A}\}$. Subsequently, the standard majority vote procedure is applied to the filtered candidates to derive the final answer $a_{sc^2} = \mathrm{mode}(\mathbb{A}_{sc})$. Recall that the model is most prone to errors when uncertain about a question, as shown in Figure 2. We can reduce inference cost by only applying the self-check filtering selectively to questions of which the certainty score $\mathrm{Certainty}(q; k)$ falls below a predefined threshold $C$.

Table 1: Evaluation results on GSM8K using the chain-of-thought prompt from Wei et al. (2022b). The self-consistency with self-check filtering technique outperforms the standard one across all models. [a]Taken from Google et al. (2023).

| Model | CoT | CoT+SC@64 | CoT+SC$^2$@64 |
|---|---|---|---|
| ChatGPT | 76.3 | 83.5 | 84.0 (+0.5) |
| PaLM-2 | 80.7[a] | 91.3 | 92.7 (+1.4) |
| GPT-4 | 91.3 | 95.8 | 96.2 (+0.4) |

We assess the performance of PaLM-2, ChatGPT and GPT-4 on the GSM8K dataset using the self-consistency with self-check method, as presented in Table 1. We use a certainty threshold of $C = 2$ for GPT-4 and $C = 1$ for both PaLM-2 and ChatGPT. Compared to self-consistency baselines, the additional self-check procedure achieves 3.03%, 16.09%, and 9.52% error reduction rate for ChatGPT, PalM-2 and GPT-4 respectively, highlighting the value of critique ability.

It is noted that our primary objective of this paper is to explore the concept and attributes of critique ability, rather than advancing the state of the art. Thus, we opt to stick with the prompting-based critic model for the sake of simplicity. While fine-tuning the critic model or using critiques to supervise the policy model could potentially push the scores higher, such enhancements are not the focus of this study. We believe future work in this direction can further improve the performance.

## 6    CONCLUSION

In this work, we conduct a study exploring critique abilities of LLMs across various tasks. Evaluation results of multiple widely-used LLMs on the proposed CRITICBENCH reveal that: most LLMs find critique challenging, especially self-critique. We introduce the *self-check* method as an effective baseline to improve model performance through self-critique. Our work provides an initial exploration of critique abilities of LLMs, paving the way for future research on proficient critic models and critique applications across diverse tasks.

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
