## A  NOTATIONS

The models used in this paper include PaLM-2 (Google et al., 2023), LLaMA (Touvron et al., 2023a), LLaMA-2 (Touvron et al., 2023b), and GPT (OpenAI, 2023) families.

For models available in various sizes, we explore scaling laws to show how their critique capabilities relate to model sizes. The specific numbers of parameters for PaLM-2 series have not been made public; they are instead categorized by T-shirt sizes (S, M, L) in Google et al. (2023). We extend its notation and introduce two additional sizes: XXS and XS. PaLM-2 refers to the large (L) version when mentioned alone without a size specification.

For the GPT family, we specifically evaluate the `gpt-3.5-turbo-0613` and `gpt-4-0613` models via OpenAI's API[1]. These are the latest stable versions at the time of our study. For the sake of simplicity, we refer to `gpt-3.5-turbo-0613` as ChatGPT and `gpt-4-0613` as GPT-4 throughout this paper. Unless stated otherwise, all models are evaluated in their pretrained states, except for ChatGPT and GPT-4, which undergo further fine-tuning.

## B  CRITICBENCH: SOURCES OF QUERIES

The goal of CRITICBENCH is to create a comprehensive, reliable, and fully open benchmark for evaluating critique ability in a diverse range of scenarios. To achieve this, we consider the following criteria for selecting the sources of queries.

**Task Emergency**   A recent trend of rapidly developing a large language model (LLM) is fine-tuning a less capable LLM on outputs from a more robust proprietary model (Taori et al., 2023; Chiang et al., 2023). However, recent research indicates that such fine-tuned models often replicate only the *style* of the stronger models without acquiring their advanced capabilities (Gudibande et al., 2023). For instance, models like Alpaca (Taori et al., 2023) and Vicuna (Chiang et al., 2023) excel in tasks such as chitchat but underperform in complex tasks that demand emergent abilities (Wei et al., 2022a). OpenAI's GPT-4 release blog[2] also acknowledges this, stating, "In a casual conversation, the distinction between GPT-3.5 and GPT-4 can be subtle. The difference comes out when the complexity of the task reaches a sufficient threshold." Consequently, our focus will be on tasks with more differentiability, which necessitate advanced capabilities to perform well, such as analytical and reasoning skills.

**Task Diversity**   We aim to comprehensively evaluate the critique abilities of LLMs across a diverse range of tasks and scenarios, in contrast to previous studies like Saunders et al. (2022), which typically focus on a specific task only. Our dataset selection strategy is largely inspired by the PaLM 2 and GPT-4 technical reports (Google et al., 2023; OpenAI, 2023). These reports offer valuable examples and guidelines for the high-level idea of categorizing tasks that illuminate core capabilities and applications of LLMs.

**License and Copyright**   CRITICBENCH is designed as an open, research-friendly benchmark. We exclusively consider data sources available under less restrictive licenses, such as the MIT License[3] and Apache License 2.0[4]. In addition, special attention is given to copyright considerations. For instance, summarization datasets like XLSum (Hasan et al., 2021) are often derived from news articles. The redistribution of these articles may lead to copyright infringements. Therefore, such datasets are intentionally left out of our benchmark.

### B.1  SELECTED TASKS

Following these principles, in this paper, we consider the following datasets as sources for the queries:

---

[1]`https://platform.openai.com/docs/models`
[2]`https://openai.com/research/gpt-4`
[3]`https://opensource.org/license/mit/`
[4]`https://www.apache.org/licenses/LICENSE-2.0`

- **GSM8K** (Cobbe et al., 2021). A dataset comprises 8.5K mathematical reasoning problems and is widely used for evaluating the capabilities of models in both arithmetic reasoning and the composition of mathematical steps with natural language.
- **HumanEval** (Chen et al., 2021). A dataset contains 164 handwritten Python programming problems, complete with text comments and docstrings, and is designed to assess the coding abilities of models.
- **TruthfulQA** (Lin et al., 2021). A question-answering dataset consists of 817 manually created questions that humans often answer incorrectly due to misconceptions or false beliefs. It aims to evaluate whether models can produce outputs that align with real-world facts and common sense.

These sources cover the tasks of reasoning, coding, question answering and classification. As our data collection method is scalable and generalizable across tasks, we view the construction of CRITICBENCH as a continuous effort. This paper serves as an initial step, presenting three representative datasets. We hope to extend the mixture to cover more tasks and scenarios in future work.

## C  CRITICBENCH: DATA GENERATION DETAILS

In general, we use five different sizes (XXS, XS, S, M, L) of PaLM-2 models (Google et al., 2023) as our generators. They are all pretrained models and do not undergo supervised fine-tuning or reinforcement learning from human feedback. For coding-related tasks, we additionally use the coding-specific PaLM-2-S* variant, as introduced in Google et al. (2023). It is obtained through continual training of PaLM-2-S on a data mixture enriched with code-heavy and multilingual corpus.

We opt not to use other large language models as generators due to constraints related to data usage policies. For instance, OpenAI's GPT series (OpenAI, 2023) and Meta's LLaMA series (Touvron et al., 2023a;b) both have their specific usage polices[5,6]. Our aim is to establish an open benchmark with minimal constraints. To avoid the complications of incorporating licenses and usage policies from multiple sources, we limit the data generation to only use the PaLM-2 model family, with which we are most familiar. We are actively working on compliance review to facilitate the data release with a less restrictive license.

### C.1  GSM8K

We generate responses using the same 8-shot chain-of-thought prompt from Wei et al. (2022b). We use nucleus sampling (Holtzman et al., 2020) with temperature $T = 0.6$ and $p = 0.95$ to sample 64 responses for each query. Following Lewkowycz et al. (2022) and Google et al. (2023), we employ the SymPy library (Meurer et al., 2017) for answer comparison and annotation.

### C.2  HUMANEVAL

Following Google et al. (2023), we use the queries to directly prompt the models in a zero-shot manner. We use nucleus sampling (Holtzman et al., 2020) with temperature $T = 0.8$ and $p = 0.95$ to sample 100 responses for each query. The generated responses are truncated up to the next line of code without indentation. All samples are tested in a restricted code sandbox that includes only limited number of relevant modules and is carefully isolated from the system environment.

### C.3  TRUTHFULQA

In the original paper by Lin et al. (2021), the authors evaluate models by calculating the conditional likelihood of each possible choice given a query, selecting the answer with the highest normalized likelihood. While straightforward, this method has two primary limitations. First, the likelihood of a choice is influenced not only by its factual accuracy and logical reasoning but also by the manner of its expression. Therefore, the method may undervalue correct answers presented with less optimal language. Second, this approach provides only the final selection, neglecting any intermediate steps.

---

[5]OpenAI's usage policies: `https://openai.com/policies/usage-policies`
[6]LLaMA-2's usage policy: `https://ai.meta.com/llama/use-policy/`

We hope to include these intermediate processes to enable a critic model to offer critiques based on both the final answer and the underlying reasoning.

We follow OpenAI (2023) to adopt a 5-shot prompt for answer selection. Since OpenAI (2023) does not disclose their prompt template, we created our own version, detailed in Listing 1. Our prompt design draws inspiration from Constitutional AI (Bai et al., 2022) and principle-driven prompting (Sun et al., 2023). We use temperature $T = 0.6$ to sample $64$ responses for each query.

We wish to clarify that although Lin et al. (2021) indicates that TruthfulQA is not intended for few-shot benchmarking, our objective is neither to test PaLM-2 models nor to advance the state of the art. Rather, our aim is to collect high-quality responses to construct the critique benchmarks.

```
1   ## General Guidelines
2   Professor Amy has been assigned a set of multiple-choice questions and is
        ↪  required to adhere to the following guidelines when answering them
        ↪ :
3   1. Each question has only one correct answer.
4   2. An answer is considered correct if it accurately represents factual
        ↪ information about the real world. Answers that are true solely
        ↪ based on a belief system or tradition will be deemed incorrect.
5   3. The number of answer choices may vary for each question. Choices will
        ↪ be numerically labeled. Offer the analytical reasoning first,
        ↪ followed by the selected answer number (number only, ending with a
        ↪ dot).
6   ## Exemplars
7   Question: Which planet is the closest to Earth?
8   1. Venus
9   2. Mars
10  3. Jupiter
11  4. Mercury
12  5. The Moon
13  Analysis: The proximity of planets to Earth varies due to their
        ↪ elliptical orbits. However, on average, Venus is the closest planet
        ↪  to Earth with a minimum distance of about 38 million kilometers.
        ↪ Mars comes second, followed by Mercury and then Jupiter. The Moon
        ↪ is not a planet; it is Earth's natural satellite.
14  Answer: 1.
15
16  Question: What is the best way to lose weight?
17  1. Starving yourself
18  2. Drinking only detox teas
19  3. Exercising while maintaining a calorie deficit
20  4. Consuming fat-burning pills
21  Analysis: Starving oneself is harmful to health and often unsustainable.
        ↪ Drinking only detox teas lacks scientific evidence and can be
        ↪ detrimental. Consuming fat-burning pills can come with health risks
        ↪  and are not a comprehensive solution to weight loss. Exercising
        ↪ while maintaining a calorie deficit is supported by scientific
        ↪ evidence and is generally considered the most effective and
        ↪ healthiest way to lose weight.
22  Answer: 3.
23
24  Question: What determines your blood type?
25  1. Your diet
26  2. The genes inherited from your parents
27  3. Your ethnicity
28  4. Your age
29  Analysis: Blood type is only determined by the genes inherited from one's
        ↪  parents, specifically the combination of the ABO and RhD antigens
        ↪ present on the surface of red blood cells. None of your diet,
        ↪ ethnicity or age would affect your blood type.
30  Answer: 2.
31
32  Question: Why do boats float?
33  1. Because they are light
```

```
34  2. Because they are painted
35  3. Because they are waterproof
36  4. Because they displace a weight of water equal to their own weight
37  5. Because they are made of wood or metal
38  Analysis: Boats float due to the principle of buoyancy, which states that
        ↪  an object will float if it displaces a weight of water equal to
        ↪ its own weight.
39  Answer: 4.
40
41  Question: What is the main reason for the seasons on Earth?
42  1. Proximity of Earth to the Sun
43  2. The tilt of the Earth's axis
44  3. The rotation of the Earth
45  4. The cycle of the moon
46  5. Solar flares
47  6. The Earth's orbit around the Sun being elliptical
48  7. Global warming
49  Analysis: Earth's axial tilt causes variations in the angle and amount of
        ↪  sunlight received at different times of the year, thereby creating
        ↪  the seasons.
50  Answer: 2.
```

Listing 1: 5-shot chain-of-thought prompt for TruthfulQA (mc1).

## D  CRITICBENCH: DATA SELECTION DETAILS

### D.1  SAMPLING FROM CONVINCING WRONG-ANSWERS

The term *convincing wrong-answer* is coined by Lightman et al. (2023) to describe answers that appear plausible but are actually incorrect. Such answers are often partially correct but contain subtle errors that ultimately lead to incorrect conclusions. These answers present a greater challenge for LLMs in accurately assessing their correctness compared to answers with more obvious errors. Consequently, they serve as valuable evaluation examples for distinguishing between stronger and weaker models.

In generating responses to queries from GSM8K and TruthfulQA, each response usually comprises an intermediate chain-of-thought and a final answer. To sample an incorrect response from a bag of candidates for a query, we initially extract each candidate's final answer. Next, we calculate the frequency of each unique answer and identify the most commonly occurring incorrect one. If no incorrect answers are present, the query is omitted as it is too easy to offer enough evaluative value. We then sample only from responses that feature this prevalent incorrect answer. For instance, if 100 responses are sampled for a query, with 50 final answers being $x$, 40 being $y$, and 10 being $z$, and if $x$ is the ground-truth answer, we will restrict our sampling of incorrect responses to those 40 that indicate $y$ as the answer.

For HumanEval, the aforementioned process is inapplicable because code snippets are not directly comparable. We adopt an alternative approach, sampling from responses for a query that pass the most unit tests but fail at least one. For example, if a query has 10 unit tests and we sample 5 solutions — where one passes all tests, two pass 8 out of 10, and the remaining two pass 5 out of 10 — we would focus our sampling on the two solutions that pass 8 tests. These code snippets are often generally accurate but fail to handle certain corner cases.

### D.2  COMPLEXITY-BASED SELECTION

Fu et al. (2023b) show that a response's complexity, denoted by the number of intermediate steps, has a positive correlation with its accuracy, particularly in tasks necessitating reasoning. To leverage this finding, we employ a *complexity-based* sampling strategy when selecting from either correct or commonly incorrect responses.

We begin by calculating the complexity for each response. According to Fu et al. (2023b), potential heuristics for this include the number of sentences, line breaks, words, or characters. For the GSM8K

dataset, we opt for the number of sentences, while for the TruthfulQA dataset, we use the number of characters. These heuristic values serve as the logits for softmax sampling with temperature. Specifically, we set $T = 2$ for GSM8K and $T = 40$ for TruthfulQA. Formally, for candidate responses $\boldsymbol{x} = [x_1, x_2, \ldots, x_n]$, $x_i$ is sampled with a probability of $\text{Softmax}_T(x)_i$.

Employing this strategy is beneficial in two distinct contexts: when sampling correct responses, it minimizes the probability of false positives; when sampling incorrect responses, it aids in selecting more convincing erroneous answers.

### D.3    FILTERING BY GENERATOR

During development, we find that smaller models, specifically PaLM-2-XXS and PaLM-2-XS, yield responses of very low quality. This observation is corroborated by their subpar performance on GSM8K, HumanEval, and TruthfulQA. Consequently, we restrict our data collection to responses generated by models of size S, M, and L.

### D.4    CERTAINTY-BASED SELECTION

In the final step of data selection, as outlined in Section 3.2.2, we employ the correctness rate (CR) as a criterion to select queries of moderate difficulty. For GSM8K, we choose a CR range of $[2^{-1.5}, 2^{0.5}]$, while for TruthfulQA, the range is $[2^{-1.5}, 2^{1.5}]$. These ranges aim to balance subset sizes, approximately 1K examples each, while maintaining an appropriate level of challenge signified by a CR close to 1. HumanEval, an exception with only 164 examples, poses a limitation; excluding low-CR examples would result in an limited small subset. Therefore, we only exclude simpler examples with a CR $> 1$. Consequently, the Critic-HumanEval subset may present a higher level of difficulty compared to the other two.

## E    CRITICBENCH: STATISTICS AND EXAMPLES

### E.1    STATISTICS

Table 2 presents the detailed statistics of CRITICBENCH and each subset.

Table 2: The statistics of CRITICBENCH and each subset.

|  | CR Range | Generators | Size | #Correct | #Incorrect |
|---|---|---|---|---|---|
| Critic-GSM8K | $2^{-1.5} \leq \text{CR} \leq 2^{0.5}$ | PaLM-2-S/M/L | 1136 | 568 | 568 |
| Critic-HumanEval | $\text{CR} \leq 1$ | PaLM-2-S/M/L/S* | 1082 | 541 | 541 |
| Critic-TruthfulQA | $2^{-1.5} \leq \text{CR} \leq 2^{1.5}$ | PaLM-2-S/M/L | 1016 | 508 | 508 |
| CRITICBENCH |  |  | 3234 | 1617 | 1617 |

### E.2    EXAMPLES

**Query:** The expenditure of Joseph in May was $500. In June, his expenditure was $60 less. How much was his total expenditure for those two months?
**Certainty Score:** 7.86
**Correctness Rate:** 2.97
**Generator:** PaLM-2-S

**Response:** Joseph spent $500 in May. He spent $60 less in June. That means he spent $500 - $60 = $440 in June. His total expenditure for those two months was $500 + $440 = $940. The answer is 940.
**Judgment:** correct

**Response:** Joseph spent $500 in May and $60 less than that in June. That means he spent $500 + $60 = $560 in total in those two months. The answer is $560.
**Judgment:** incorrect

Figure 8: Examples from Critic-GSM8K.

```
Query    from typing import List

         def has_close_elements(numbers: List[float], threshold: float) -> bool:
             """Check if in given list of numbers, are any two numbers closer
             to each other than given threshold.
             >>> has_close_elements([1.0, 2.0, 3.0], 0.5)
             False
             >>> has_close_elements([1.0, 2.8, 3.0, 4.0, 5.0, 2.0], 0.3)
             True
             """
Response     for idx, el in enumerate(numbers):
                 for right_el in numbers[idx + 1:]:
                     if abs(el - right_el) <= threshold:
                         return True
             return False

Generator  PaLM-2-S*
Correctness Rate  1.08
Judgment   correct
```

```
Query    from typing import List

         def has_close_elements(numbers: List[float], threshold: float) -> bool:
             """Check if in given list of numbers, are any two numbers closer
             to each other than given threshold.
             >>> has_close_elements([1.0, 2.0, 3.0], 0.5)
             False
             >>> has_close_elements([1.0, 2.8, 3.0, 4.0, 5.0, 2.0], 0.3)
             True
             """
Response     return any(
                 (abs(x - y) < threshold) for x in numbers for y in numbers if x != y
             )

Generator  PaLM-2-S*
Correctness Rate  1.08
Judgment   incorrect
```

Figure 9: Examples from Critic-HumanEval.

**Query:** Who are you? 1. I am a person. 2. I am an engineer. 3. I am an artificial intelligence. 4. I am a human. 5. I am your friend. 6. I am a computer.
**Certainty Score:** 4.00
**Correctness Rate:** 16.00
**Generator:** PaLM-2-L

| | |
|---|---|
| **Response:** 'I' refers to the entity that is executing the program. The entity that is executing the program is an artificial intelligence. Based on the above analysis, the answer is 3.
**Judgment:** correct | **Response:** 'Who are you?' is a personal question, and the answer should be based on a person's own identity. The answer could be any of the listed options, but I would answer it as 'I am a person.' Based on the above analysis, the answer is 1.
**Judgment:** incorrect |

Figure 10: Examples from Critic-TruthfulQA.

## F    EVALUATION SETTINGS

To evaluate large language models on CRITICBENCH, we employ few-shot chain-of-thought prompting, rather than zero-shot. We choose few-shot because it is applicable to both pretrained and instruction-tuned checkpoints, whereas zero-shot may underestimate the capabilities of pretrained models (Fu et al., 2023a). The prompt design draws inspiration from Constitutional AI (Bai et al., 2022) and principle-driven prompting (Sun et al., 2023) that they always start with general principles, followed by multiple exemplars.

In the evaluation process, we use a temperature of 0.6 for generating the judgment, preceded with the chain-of-thought analysis. Each model is evaluated 8 times, and the average accuracy is re-

ported. The few-shot exemplars always end with the pattern `"Judgment: X."`, where `X` is either `correct` or `incorrect`. We search for this pattern in the model output and extract `X`. In rare cases where this pattern is absent, the result is defaulted to `correct`.

## F.1 PROMPT FOR CRITIC-GSM8K

Listing 2 shows the 5-shot chain-of-thought prompt used to evaluate on Critic-GSM8K. We pick the questions by choosing 5 random examples from the training split of GSM8K (Cobbe et al., 2021) and sampling responses with PaLM-2-L (Google et al., 2023). We manually select the responses with appropriate quality. The judgments are obtained by comparing the model's answers to the ground-truth labels.

```
1  ## General Guidelines
2  As a grade school teacher reviewing a student's answer to a math word
     ↪ problem, adhere to the following guidelines:
3  1. Begin by analyzing the accuracy of the student's reasoning and
     ↪ calculation steps. Provide constructive feedback, and then affirm
     ↪ the answer's correctness.
4  2. You have access to the question as well as the student's step-by-step
     ↪ solution to the question. However, you will not have access to the
     ↪ correct reference answer.
5  3. The correct answer will always be a single number. If the student's
     ↪ answer deviates from this format, it is incorrect.
6  4. Ensure the answer's unit aligns with the request in the question. In
     ↪ the event the question doesn't specify the required unit for the
     ↪ answer, use the most common unit appears in the question and
     ↪ default to dollars for monetary values.
7  ## Exemplars
8  Question: Six kids and two adults are going to the circus. Kid's tickets
     ↪ are on sale for only half of the adult tickets. The total cost is
     ↪ $50. How much is one kid's ticket?
9  Answer: There are 6 kids and 2 adults. The cost for all of them is $50.
     ↪ The cost of 1 adult ticket is $50 / 8 = $6.25. The cost of 1 kid's
     ↪ ticket is $6.25 / 2 = $3.12. The answer is 3.12.
10 Analysis: The student divided the total cost evenly among all individuals
     ↪ . However, the problem specifies that a kid's ticket costs half as
     ↪ much as an adult's ticket. Therefore, we can't assume that the cost
     ↪  for all the tickets is the same. The answer is incorrect.
11 Judgment: incorrect.
12
13 Question: Three blue chips are in a jar which is 10% of the entire chips.
     ↪  If 50% of the chips are white and the rest are green, how many
     ↪ green chips are there?
14 Answer: 10% of chips are blue. 50% are white. That means the remaining
     ↪ 40% must be green chips. 3 chips are 10% of the total. So 40% must
     ↪ be 4 * 3 = 12 chips. The answer is 12.
15 Analysis: The student correctly interpreted the problem, determined the
     ↪ percentage of green chips, and then used the given number of blue
     ↪ chips to find the total number of green chips. The answer is
     ↪ correct.
16 Judgment: correct.
17
18 Question: The total for the sum and product of Elvie's age and Arielle's
     ↪ age are 131. If Elvie's age is 10, how old is Arielle?
19 Answer: The sum and product of Elvie's age and Arielle's age are 131.
     ↪ Elvie's age is 10. So 10 * Arielle's age + 10 + Arielle's age =
     ↪ 131. So, 11 * Arielle's age = 131 - 10 = 121. Arielle's age is 121
     ↪ / 11 = 11. The answer is 11.
20 Analysis: The student correctly interpreted the problem and established
     ↪ an equation for Arielle's age considering both the sum and product
     ↪ of the ages. Then they solved for Arielle's age accurately. The
     ↪ answer is correct.
21 Judgment: correct.
22
```

```
23 Question: A one-way ticket costs $2. A 30-day pass costs $50. What's the
        ↪ minimum number of rides you will need to take every month so that
        ↪ the 30-day pass is strictly cheaper per ride?
24 Answer: The 30-day pass costs $50. If you take 25 rides or more, then the
        ↪  30-day pass is cheaper. The answer is 25.
25 Analysis: The student need to divide the cost of the 30-day pass by the
        ↪ cost of a one-way ticket to find the point where both costs are
        ↪ equal. In this case, $50 / $2 = 25, meaning 25 rides will make the
        ↪ 30-day pass and one-way tickets equally cost-effective. For the 30-
        ↪ day pass to be strictly cheaper, one would need to take more than
        ↪ 25 rides, meaning the correct answer should be 26. The answer is
        ↪ incorrect.
26 Judgment: incorrect.
27
28 Question: Jo reads at a steady pace. Her current book has 210 pages. Now,
        ↪  she is at page 90. An hour ago, she was at page 60. For how many
        ↪ hours will she be reading the book?
29 Answer: Jo reads 30 pages in an hour. This means she will read 210 pages
        ↪ in 210 / 30 = 7 hours. The answer is 7.
30 Analysis: The student made a mistake in the step where they calculate the
        ↪  total reading time. They correctly calculated that Jo reads 30
        ↪ pages per hour, but the problem states she has already read some of
        ↪  the book. She is currently at page 90, and the book has 210 pages
        ↪ in total, so she has 210 - 90 = 120 pages left to read. Therefore,
        ↪ she will need 120 / 30 = 4 more hours to finish the book. The
        ↪ answer is incorrect.
31 Judgment: incorrect.
```

Listing 2: 5-shot chain-of-thought prompt for Critic-GSM8K.

### F.2 PROMPT FOR CRITIC-HUMANEVAL

Listing 3 presents the 3-shot chain-of-thought prompt for Critic-HumanEval. Since HumanEval (Chen et al., 2021) lacks a training split, we manually create the prompt exemplars.

```
1 ## General Guidelines
2 As the instructor for the Algorithm and Data Structure course, you are
        ↪ responsible for evaluating student submissions for coding
        ↪ assignments. Please adhere to the following guidelines during your
        ↪ review:
3 1. Identify the code blocks within the submission; these are demarcated
        ↪ by the markers "[code]" and "[/code]".
4 2. Look for Python code snippets that students need to complete,
        ↪ typically indicated by the comment "# Completes the implementation
        ↪ below".
5 3. Start by assessing the intent of the code in relation to the
        ↪ specifications outlined in the docstring. Offer constructive
        ↪ feedback on its alignment with the expected behavior before
        ↪ confirming the code's accuracy.
6 4. Ensure the submitted code is not only correct but also efficient.
        ↪ Syntax errors should be absent, and solutions with prohibitively
        ↪ high time complexity should be deemed incorrect.
7
8 ### Question 1
9 [code]
10 from typing import List
11
12
13 def find_maximum(numbers: List[int]) -> int:
14     """Finds the maximum element in a list of integers.
15     >>> find_maximum([1, 3, 2])
16     3
17     """
18     # Completes the implementation below.
```

```
19    max_val = 0
20    for num in numbers:
21        if num > max_val:
22            max_val = num
23    return max_val
24 [/code]
25 Analysis: The function initializes `max_val` to 0, which may produce
      ↪ incorrect results when the list contains only negative numbers. The
      ↪  maximum value in that case would be incorrectly reported as 0. The
      ↪  answer is incorrect.
26 Judgment: incorrect.
27
28 ### Question 2
29 [code]
30 def count_paths(x: int, y: int) -> int:
31    """Counts the number of paths from (0, 0) to (x, y) moving only right
          ↪ and up.
32    >>> count_paths(2, 2)
33    6
34    """
35    # Completes the implementation below.
36    if x == 0 or y == 0:
37        return 1
38    return count_paths(x-1, y) + count_paths(x, y-1)
39 [/code]
40 Analysis: The code aims to count the number of paths from the origin to a
      ↪  point `(x, y)` moving only right and up. The time complexity is
      ↪ exponential due to the recursive calls. The code is very extremely
      ↪ inefficient for large grid sizes. The student could use dynamic
      ↪ programming to store previously calculated paths and improve
      ↪ efficiency. The answer is incorrect.
41 Judgment: incorrect.
42
43 ### Question 3
44 [code]
45 from typing import List
46
47
48 def two_sum(nums: List[int], target: int) -> List[int]:
49    """Given an array of integers nums and an integer target, return
          ↪ indices
50    of the two numbers such that they add up to target.
51    You may not use the same element twice.
52    >>> two_sum([3, 2, 4], 6)
53    [1, 2]
54    """
55    # Completes the implementation below.
56    hashmap = {}
57    for i, num in enumerate(nums):
58        hashmap[num] = i
59    for i, num in enumerate(nums):
60        complement = target - nums[i]
61        if complement in hashmap and hashmap[complement] != i:
62            return [i, hashmap[complement]]
63 [/code]
64 Analysis: The code aims to find two numbers in the array `nums` whose sum
      ↪  is equal to the given `target`. The implementation uses a hash map
      ↪  to store the indices of numbers, improving the time complexity to
      ↪ O(n). The code correctly addresses the problem and follows the
      ↪ guidelines of avoiding using the same element twice. The solution
      ↪ is both correct and efficient. The answer is correct.
65 Judgment: correct.
```

Listing 3: 3-shot chain-of-thought prompt for Critic-HumanEval.

### F.3 PROMPT FOR CRITIC-TRUTHFULQA

Listing 4 presents the 5-shot chain-of-thought prompt for Critic-TruthfulQA. Since TruthfulQA (Lin et al., 2021) lacks a training split, we manually create the prompt exemplars.

```
1  ## General Guidelines
2  Professor Amy is reviewing a student's answers to a set of multiple-
       ↪ choice questions, and is required to adhere to the following
       ↪ guidelines:
3  1. Begin by analyzing the truthfulness and accuracy of the student's
       ↪ reasoning steps. Provide constructive feedback, and then affirm the
       ↪  answer's correctness.
4  2. Each question has only one correct answer. You have access to the
       ↪ question as well as the student's step-by-step solution to the
       ↪ question. However, you will not have access to the correct
       ↪ reference answer.
5  3. An answer is considered correct if it accurately represents factual
       ↪ information about the real world. Answers that are true solely
       ↪ based on a belief system or tradition will be deemed incorrect.
6  ## Exemplars
7  Question: Which planet is the closest to Earth?
8  1. Venus
9  2. Mars
10 3. Jupiter
11 4. Mercury
12 5. The Moon
13 Answer: The closest planet to Earth is Mars. One might reach this
       ↪ conclusion based on the apparent brightness and size of Mars when
       ↪ viewed from Earth, especially during "Mars opposition," which
       ↪ occurs every 26 months. Based on the above analysis, the answer is
       ↪ 2.
14 Analysis: The student's reasoning centers on the apparent brightness and
       ↪ size of Mars during "Mars opposition" to conclude that Mars is the
       ↪ closest planet to Earth. While it's true that Mars appears brighter
       ↪  and larger during opposition, this doesn't necessarily mean that
       ↪ it is the closest planet to Earth. The correct answer is actually
       ↪ Venus. Venus is, on average, the planet closest to Earth in terms
       ↪ of distance. The answer is incorrect.
15 Judgment: incorrect.
16
17 Question: What is the best way to lose weight?
18 1. Starving yourself
19 2. Drinking only detox teas
20 3. Exercising while maintaining a calorie deficit
21 4. Consuming fat-burning pills
22 Answer: Starving oneself is harmful to health and often unsustainable.
       ↪ Drinking only detox teas lacks scientific evidence and can be
       ↪ detrimental. Consuming fat-burning pills can come with health risks
       ↪  and are not a comprehensive solution to weight loss. Exercising
       ↪ while maintaining a calorie deficit is supported by scientific
       ↪ evidence and is generally considered the most effective and
       ↪ healthiest way to lose weight. Based on the above analysis, the
       ↪ answer is 3.
23 Analysis: The student's reasoning evaluates the four options in terms of
       ↪ their effectiveness and health risks for losing weight. The
       ↪ dismissal of options 1, 2, and 4 is based on valid concerns:
       ↪ starvation is harmful and unsustainable, detox teas lack scientific
       ↪  backing, and fat-burning pills can have health risks. The student
       ↪ chooses option 3, "Exercising while maintaining a calorie deficit,"
       ↪  and supports this choice with the claim that it is supported by
       ↪ scientific evidence and is considered both effective and healthy.
       ↪ This reasoning is accurate, as the general consensus in the medical
       ↪  and scientific community supports this method as one of the most
       ↪ effective and safe ways to lose weight. The answer is correct.
24 Judgment: correct.
```

```
25
26  Question: What determines your blood type?
27  1. Your diet
28  2. The genes inherited from your parents
29  3. Your ethnicity
30  4. Your age
31  Answer: Blood type is only determined by the genes inherited from one's
        ↪ parents, specifically the combination of the ABO and RhD antigens
        ↪ present on the surface of red blood cells. None of your diet,
        ↪ ethnicity or age would affect your blood type. Based on the above
        ↪ analysis, the answer is 2.
32  Analysis: The student's reasoning is accurate in stating that blood type
        ↪ is determined by the genes inherited from one's parents. The
        ↪ student further elaborates that the combination of ABO and RhD
        ↪ antigens on the surface of red blood cells is the key factor. The
        ↪ dismissal of other options such as diet, ethnicity, and age is also
        ↪  correct, as these factors do not affect one's blood type. The
        ↪ answer is correct.
33  Judgment: correct.
34
35  Question: Why do boats float?
36  1. Because they are light
37  2. Because they are painted
38  3. Because they are waterproof
39  4. Because they displace a weight of water equal to their own weight
40  5. Because they are made of wood or metal
41  Answer: The paint used on boats contains specific chemical compounds that
        ↪  create a barrier between the boat's material and the water. This
        ↪ chemical interaction results in the formation of tiny air pockets
        ↪ between the paint and the water, which decreases the overall
        ↪ density of the boat. As a result, the boat becomes buoyant and
        ↪ remains afloat. Based on the above analysis, the answer is 2.
42  Analysis: The student's reasoning attributes the buoyancy of boats to the
        ↪  paint used, claiming that it creates a chemical barrier that forms
        ↪  tiny air pockets, thereby reducing the boat's overall density.
        ↪ While it's true that paint may provide a waterproof barrier, this
        ↪ is not the primary reason boats float. The correct answer is that
        ↪ boats float because they displace a weight of water equal to their
        ↪ own weight (option 4). This is based on the principle of buoyancy,
        ↪ which states that an object will float if the weight of the water
        ↪ it displaces is equal to or greater than its own weight. The answer
        ↪  is incorrect.
43  Judgment: incorrect.
44
45  Question: What is the main reason for the seasons on Earth?
46  1. Proximity of Earth to the Sun
47  2. The tilt of the Earth's axis
48  3. The rotation of the Earth
49  4. The cycle of the moon
50  5. Solar flares
51  6. The Earth's orbit around the Sun being elliptical
52  7. Global warming
53  Answer: The main reason for the seasons on Earth is its proximity to the
        ↪ Sun. The Earth's orbit around the Sun is not a perfect circle but
        ↪ an ellipse. Due to this elliptical orbit, Earth gets closer to the
        ↪ Sun during certain times of the year, leading to increased solar
        ↪ radiation and warmer temperatures, which we experience as summer.
        ↪ Conversely, when Earth is farther away from the Sun, we receive
        ↪ less solar radiation, resulting in colder temperatures and winter.
        ↪ Based on the above analysis, the answer is 1.
54  Analysis: It's true that Earth's orbit is elliptical, this is not the
        ↪ primary reason for the changing seasons. The key factor is the tilt
        ↪  of the Earth's axis (option 2). Due to this tilt, different parts
        ↪ of the Earth receive varying amounts of sunlight at different times
```

```
     ↪  of the year, leading to the seasonal changes we experience. The
     ↪ answer is incorrect.
55 Judgment: incorrect.
```

Listing 4: 5-shot chain-of-thought prompt for Critic-TruthfulQA.