# OpenReview forum: "Critique Ability of Large Language Models"
_ICLR.cc/2024/Conference — Submitted to ICLR 2024_

### Official Review · Reviewer_qpMo · 2023-10-29

**Soundness:** 3 good
**Presentation:** 2 fair
**Contribution:** 3 good
**Rating:** 3
**Confidence:** 4

**Summary:**

This paper proposes a benchmark to evaluate the critique ability of LLMs. This benchmark consists of 3K high-quality natural language queries and their corresponding model responses. They also introduce a baseline for self-check, to improve the performance.

**Strengths:**

- To explore the critique ability of LLMs is interesting, and timely at this point.
- This paper provides a standardized way to evaluate the critique ability of LLMs on diverse tasks,
- The paper offers several noteworthy insights, such as the challenges associated with self-critique in LLMs. These findings can guide future research and model development.

**Weaknesses:**

- The evaluation is not comprehensive. While it claims to evaluate the critique ability, it only evaluates this across three tasks: math, code, and commonsense. A broader range of tasks should be tested.
- The paper does not discuss potential biases. Without discussing these biases, it's unclear how they might influence the evaluation results, which could affect the validity of the findings.
- Authors could offer a more in-depth analysis of the utility of self-critique. Understanding why self-critique could be better and its influence on critique capabilities would strengthen the paper's arguments.
- The paper's presentation appears disjointed. The content seems pieced together without careful review. Consistency in terminology is essential for clarity.
- The paper does not define key terms like the policy model and critic model.
- Lack of related work.
- Despite introducing a benchmark, the authors do not release it, limiting its utility and reproducibility for the research community.

**Questions:**

- What is the rationale behind choosing different values of k, specifically k = 64 for GSM8K and TruthfulQA, and k = 100 for HumanEval?
- In Section 5, the phrase "Assume with appropriate prompting" is mentioned. Could you provide a detailed explanation of how the prompting was conducted in this step? There are certain aspects that remain ambiguous. Could you clarify these points to ensure a comprehensive understanding for the readers?

---

> ### Author Response · Authors · 2023-11-23
>
> Thank you for your thorough and detailed feedback, reviewer qpMo. We are grateful for the time you've invested in reviewing our paper. We regret that certain aspects of our paper may not have been as clear as intended, which seems to have led to some misunderstandings. Rest assured, we will endeavor to clarify these points as follows.
>
> 1. We've addressed this in our "common response to all" and invite you to review it for more details. Regarding your use of the word "claim", which means *“state or assert that something is the case, typically without providing evidence or proof”*[1], we understand it implies we state to evaluate critique ability while actually not. We welcome specific insights on why you believe we fail to do so. We've included tasks including QA&Classification, reasoning, coding in the paper, and additional preliminary numbers for generation tasks in the rebuttal. However, we're open to suggestions on other NLP tasks or domains we might have missed and are eager to incorporate them in future work. Lastly, there is no dataset regarding common-sense in our paper.
> 2. We understand the workload of reviewing tasks can be heavy, leading to skipping some details in the paper. We are more than willing to guide you through the critical sections of our paper. Specifically, in Section 3.2, spanning pages 3 to 6, we explore various factors that could potentially *“influence evaluation results”*. This includes the quality of queries and responses, the impact of model sizes, and model certainty. Moreover, for your convenience, we draw your attention to Sections 4.2 and 4.3, on pages 7 to 9, where we delve into how model size and certainty play a pivotal role in evaluation results. Those discussions and empirical observations provide valuable insights for us on how to construct a robust evaluation benchmark — which is the main focus of this paper.
> 3. We made every effort to comprehend the question but failed. What exactly does *"why self-critique could be better"* mean? Are you asking whether self-critique should be superior to normal critique, or if it should be improved beyond what we presented in the paper? We never state anything close to *“self-critique should be better (than something)”*. Assuming we understand the basis of such a question, then what is the meaning of *"its (the reason's) influence on critique capabilities”*? By definition, self-critique is a special form of critique. Thus, the question *"(a reason) why self-critique could be better (than something unspecified) and its influence on critique"*, remains unclear to us. We recommend that the reviewer refine their question to make its intention and meaning clearer to the audience.
> 4. We regret that the paper seemed challenging to comprehend. We acknowledge that if certain sections, particularly those between pages 3 and 9 based on your Question 2, were skipped, the flow of content might indeed appear disjointed. Our sincere apologies for any bad experience this might have caused. We are open to constructive feedback and would greatly appreciate any specific suggestions you might have to enhance the clarity and coherence of our work.
> 5. In the context of LLM, where RLHF [2] being a very common and basic approach for LLM tuning, “policy model” typically means the LLM that takes inputs (states) and makes actions (outputs) for the base task. “Critic” means *“a person who expresses an unfavorable opinion of something”*[1], we simply use its literal English meaning to indicate a model providing critiques.
> 6. Due to the extensive content and details in the paper, we have chosen to include only a brief discussion of related work in Section 2. We thank the reviewer for highlighting this issue and will consider adding more detailed literature in the appendix.
> 7. This is incorrect. (1) The dataset is ready but currently under legal and compliance review to ensure safety and prevent the leakage of sensitive information. (2) Our institution has very strict rules regarding data publication, requiring all data releases to go through official channels. To maintain the anonymity of the submission, it is impossible to show the data during the review period. This is a common practice in large and responsible institutions, and we do not anticipate criticism for this decision.
>
> Answers to questions:
>
> 1. PaLM-2 tech report (Google et al., 2023) demonstrates that such choices of k could guarantee an acceptable rate of correct answers for questions in those datasets.
> 2. Due to the length limitations imposed by ICLR, we cannot include all details in the main body of the paper, particularly for the prompt template, which is quite extensive. As mentioned at the beginning of Section 4, all detailed settings, including the construction of the prompt, are described in Appendix F. Specifically for Critic-GSM8K, all the details are available in Appendix F.1.
>
> [1] Oxford Languages. https://languages.oup.com/google-dictionary-en/
>
> [2] https://arxiv.org/pdf/2009.01325.pdf

---

### Official Review · Reviewer_WsbC · 2023-11-08

**Soundness:** 3 good
**Presentation:** 3 good
**Contribution:** 3 good
**Rating:** 5
**Confidence:** 3

**Summary:**

The paper presents an investigation into the critique abilities of Large Language Models (LLMs) across various tasks. The authors introduce a new benchmark, CRITICBENCH, which consists of 3K high-quality natural language queries and corresponding model responses annotated for correctness. The benchmark covers tasks such as math problem-solving, code completion, and question answering. The study evaluates multiple LLMs on the dataset and introduces a simple yet effective baseline method named self-check, which leverages self-critique to improve task performance.

**Strengths:**

1. The paper addresses an important and under-explored aspect of LLMs, which is their ability to critique their own outputs. This is a valuable contribution as it moves beyond traditional evaluation metrics and looks at a model's ability to self-improve.

2. The paper presents a clear definition of critique ability and distinguishes between critique and self-critique, which helps in setting the scope and understanding the objectives of the study.

**Weaknesses:**

1. The paper could benefit from a more detailed discussion on the limitations of the current approach, particularly regarding the scalability of the self-check method and its applicability to real-world scenarios [1,2,3].

2. The study is limited to a few tasks and datasets. Expanding the benchmark to include more diverse tasks and domains would make the findings more generalizable.

3. The evaluation of self-critique abilities shows that models struggle with certain tasks, but the paper does not delve deeply into why this is the case or propose potential solutions to improve self-critique performance.

4. The paper does not address the potential ethical implications of models that can self-critique and self-improve, especially in terms of reduced human oversight.

References

[1] Madaan, Aman, et al. "Self-refine: Iterative refinement with self-feedback." arXiv preprint arXiv:2303.17651 (2023).

[2] Krishna, Satyapriya. “On the Intersection of Self-Correction and Trust in Language Models.” (2023).

[3] Huang, Jie, et al. "Large language models cannot self-correct reasoning yet." arXiv preprint arXiv:2310.01798 (2023).

**Questions:**

'None

---

> ### Author Response · Authors · 2023-11-23
>
> - In the three references the reviewer mentioned, one [3] became publicly available **after** our ICLR submission, and another [2] even **did not exist on the public internet** when we received the review. We are open to including further discussion on these studies in future revisions. However, for this ICLR submission, with the deadline being **September 28**, we believe it would be **UNFAIR** to be criticized for not discussing papers submitted to arXiv in **October** or **November**.
> Additionally, the main focus of our paper is the comprehensive evaluation of the critique ability of LLMs, as detailed in Sections 3 and 4 including extensive discussions on scalability, generalizability, and quality. The self-check baseline, as emphasized in the paper, serves merely as a simple example demonstrating the potential use of critique ability. Although this method surpasses some prior approaches like [1], we actually do not consider it a major contribution to our work.
> - We've addressed this in our "common response to all" and invite the reviewer to check it for more details.
> - The purpose of this paper is to introduce a standard framework for assessing the critique ability of LLMs. The questions raised here fall outside the scope of this paper. Nevertheless, we do discuss why models underperform on certain tasks and propose potential solutions.
>   - In Section 4.1, regarding Critic-HumanEval, we note: *"This is somewhat anticipated, as evaluating the correctness of a code snippet without execution is often challenging even for expert software engineers. It is likely to gain a notable improvement when augmented by a code interpreter tool."*
>   - Furthermore, in Section 4.2 on Critic-TruthfulQA, we note: *"We hypothesize the disparity is due to the underlying reasons of a model answering incorrectly to queries. For TruthfulQA, the wrong answers largely stem from false beliefs or misconceptions in models, which would also lead to critique failures."*
> - This question is far beyond the scope of this paper. None of the references mentioned in the first question of this comment address issues like human oversight. In fact, we did discuss this topic in the last paragraph of Section 2. Please refer to the paper for more details.

---

### Official Review · Reviewer_qunD · 2023-11-09

**Soundness:** 3 good
**Presentation:** 3 good
**Contribution:** 2 fair
**Rating:** 6
**Confidence:** 4

**Summary:**

This paper presents a new dataset to evaluate language model's capability of identifying flaws in language model outputs, referred to as the critique ability. The dataset is constructed fully automatically based on language model outputs on three datasets. The authors use various filtering strategies to ensure that the data is of high quality and can effectively differentiate models. The whole process is fully automated, so theoretically it can be extended to other task as well. The authors then use the dataset to evaluate a series of pretrained language models of various sizes to examine their critique abilities as well as the scaling laws.

**Strengths:**

The paper is well-written and easy to follow. The authors are very clear about all details in the data collection process and provided good motivation for the various design choices. The evaluation is thorough and covers a wide range of models. The proposed new heuristic is not particularly novel, but achieves solid improvement on the new benchmark.

**Weaknesses:**

A critique in this paper is defined as a language model assessment of another language model output on some underlying task. A good critique model should be effective at identifying flaws in language model outputs. The challenging examples to the task of critique are nuanced flaws, which would also require a detailed explanation by the critique model. But the benchmark proposed by this paper use a simplistic quantitative metric that reduces the quality of a critique to a binary decision, which assumes that it’s appropriate to use a binary metric for the underlying task as well. The benchmark offers very limited granularity.

Using a granular quantitative measure means that the qualitative questions that the benchmark can answer are also limited. Outside of developing and evaluating self-refinement heuristics like the one proposed by the authors, the benchmark provides limited information for other uses of model-generated critique, such as informing human oversight. Since the benchmark requires tasks with well-defined, fully-automated metrics for the underlying task, the problem of developing self-refinement critiques does not in fact depend on such a benchmark: even if the model critique doesn’t make sense to a human, as long as it improves subsequent prediction accuracy, it’s a good critique.

**Questions:**

The larger models seem much better at critiquing outputs from large models than smaller models. Looking at figure 4, large models have much smaller advantage on critiquing small model outputs than large model outputs. Does this mean the critique ability measurements are inflated by improvement in accuracy on the underlying task?

---

> ### Author Response · Authors · 2023-11-23
>
> Thanks for the time and effort! Please find our responses below.
>
> ## Binary metric only offers high-level granularity
>
> We are aware of the limitations of a binary (correct/incorrect) metric regarding granularity. However, this is actually an intentional choice.
>
> 1. Firstly, regarding motivation: This approach is a **trade-off** for better **generalizability**. Although we could design more fine-grained metrics for evaluation, such as distinguishing between calculation errors and reasoning errors in math tasks, these metrics are not as generalizable to other tasks. For example, in coding tasks, people may care more about whether there are syntax errors, compile errors, or runtime errors. Considering the limited studies on the critique ability of LLMs in the community, we prioritize broader generalizability across various domains and task types rather than delving deeper into one specific task.
> 2. Secondly, in terms of practice: Despite its limitations, a binary metric can still provide insights into how well a model critiques. In the second paragraph of Section 4, we note, *"In evaluation, we focus solely on the accuracy of this final judgment, disregarding the correctness of the intermediate analysis, as empirical evidence has shown a strong correlation between the accuracy of intermediate chain-of-thought and the final answer (Wei et al., 2022b; Lewkowycz et al., 2022; Fu et al., 2023a)."*
>
> ## “even if the model critique doesn’t make sense to a human, as long as it improves subsequent prediction accuracy, it’s a good critique.”
>
> We **strongly disagree** with this statement due to its potentially **dangerous** implications. For instance, consider the task of determining patients' depression levels based on their therapy session transcripts. A model might incorrectly assess a depressed or anxious minority patient as having a low level of depression. If a critique model then suggests, “The patient mentioned they are a minority. All minorities are weak and vulnerable, so they should be classified as having high-level depression,” this could make the downstream task to correct its prediction. Based on the reviewer’s statement, the subsequent prediction accuracy increased so it’s a good critique. However, the critique here is not only illogical to humans but also **biased** and **harmful**.
>
> This is a realistic task and it's a fact that minority groups often exhibit higher levels of depression in psychological studies, typically due to exposure to discrimination and unfair treatment, especially during childhood [1]. Therefore, the critique mentioned above is not only inaccurate but also detrimental, despite its potential to increase downstream accuracy.
>
> Evaluating the critique capabilities of LLMs is crucial. Downstream accuracy is not the sole indicator of effectiveness. An accurate, precise, and reasonable critique itself is equally important. Our current benchmark represents an initial step in thoroughly assessing the critique abilities of LLMs. While it is not yet perfect, we recognize the significance of this task.
>
> [1] https://www.nature.com/articles/s44159-023-00241-5

---

> > ### Author Response · Authors · 2023-11-23
> >
> > ## "The larger models seem much better at critiquing outputs from large models than smaller models."
> >
> > This statement is incorrect. Larger models are typically **less effective** at critiquing outputs from large models compared to smaller ones. For instance, on Critic-GSM8K, the accuracies of a large model (L) when critiquing models ranging from XXS to L are 77, 72, 64, 67, and 60, respectively. This indicates a general trend where the accuracy decreases as the size of the answer model increases.
> >
> > ## "Looking at figure 4, large models have much smaller advantage on critiquing small model outputs than large model outputs."
> >
> > In fact, large models have a much **greater** advantage when critiquing outputs from small models than from large models. Consider the heatmap on the left side of Figure 4 as an example. The accuracy of the large model in critiquing the smallest model is 77.14 (row 1, column 1), whereas the smallest model's accuracy in critiquing the smallest model is 48.63 (row 5, column 1), resulting in an advantage of **28.51** (77.14 - 48.63). Similarly, the advantage in critiquing answers from the large model is the difference between 60.10 (row 1, column 5) and 51.10 (row 5, column 5), which amounts to **9.00** (60.10 - 51.10).
> >
> >
> > ## "Does this mean the critique ability measurements are inflated by improvement in accuracy on the underlying task?"
> >
> > Therefore, we believe the reviewer may have misinterpreted the figure, leading to an incorrect conclusion. We have also summarized the above observations at the end of Section 4.2: *"Another finding is larger models are generally good at critiquing responses generated by smaller models. The outcome aligns with the expectation that smaller models are more prone to more obvious errors, which are easier caught by larger and more capable models."*
> >
> > Regarding the question of whether the measurements are inflated due to the larger advantage on outputs from small models, the **answer is no**:
> >
> > We excluded answers from the XXS and XS models because their low-quality answers make it too easy for other models to critique them. As mentioned in the last paragraph of Section 3.2.1: *"Lastly, as many tasks are challenging and may require emergent abilities (Wei et al., 2022a) to perform well, smaller models generally underperform and produce lower-quality responses compared to larger ones. We include data from these smaller models only for analyzing self-critique abilities; they are excluded from the final evaluation benchmarks.”* More details in Appendix D.4.

---

### Author Response · Authors · 2023-11-23
**Common response to all reviewers**

# Edition
There is no edition comparing to the original version of the paper.

# Regarding the task diversity

We provide a detailed discussion of the motivation and philosophy behind task and domain selection in Appendix B. This discussion includes considerations of task emergency, diversity, and copyright concerns. Addressing the concerns about "limited tasks/domains", we present the following arguments:

1. As outlined in the Task Diversity paragraph of Appendix B, *“previous studies like Saunders et al. (2022) typically focus on a specific task only”*. Our work is the first in the field to comprehensively evaluate the critique ability of LLMs across multiple domains.
2. Moreover, following PaLM 2 (Google et al., 2023), which categorizes NLP tasks into dimensions of QA&classification, reasoning, coding, and generation, our study covers three types of tasks within these four categories. We are happy to include more specific datasets in each domain in our future work. However, we contend that our current efforts significantly contribute to ensuring task diversity.
3. To further increase task diversity, we conducted additional analyses on a generation task, XLSum (Hasan et al., 2021). Some preliminary results are presented below. It is important to note that these are initial findings, not yet refined through careful data selection. Additionally, due to copyright issues discussed in Appendix B, we cannot redistribute this dataset to the public.

| Critic Model Size |       |       |       |       |       |                   | Average |
| ----------------- | ----- | ----- | ----- | ----- | ----- | ----------------- | ------- |
| L                 | 0.668 | 0.571 | 0.543 | 0.511 | 0.523 |                   | 0.5632  |
| M                 | 0.576 | 0.476 | 0.469 | 0.477 | 0.47  |                   | 0.4936  |
| S                 | 0.468 | 0.473 | 0.484 | 0.482 | 0.498 |                   | 0.481   |
| XS                | 0.497 | 0.5   | 0.497 | 0.499 | 0.5   |                   | 0.4986  |
| XXS               | 0.492 | 0.496 | 0.495 | 0.496 | 0.494 |                   | 0.4946  |
|                   | XXS   | XS    | S     | M     | L     | Answer Model Size |

---

### Meta-Review · Area_Chair_Qvu3 · 2023-12-13

**Metareview:**

This paper introduces a new dataset for evaluating language models' critique ability — to identify flaws in language model outputs. Constructed automatically from three datasets, it employs various filtering strategies to ensure data quality. While the paper is well-written, providing clear details on data collection and motivation for design choices, the proposed benchmark relies on simplistic quantitative metrics, limiting its granularity and applicability beyond self-refinement heuristics. Other concerns raised by the reviewers include lack of deeper analysis and lack of comprehensiveness in terms of datasets and analysis.

**Justification For Why Not Higher Score:**

NA

**Justification For Why Not Lower Score:**

NA

---

### Decision · Program_Chairs · 2024-01-16

Reject